# Dual-Coupled CNN-GCN-Based Classification for Hyperspectral and LiDAR Data

**DOI:** 10.3390/s22155735

**Published:** 2022-07-31

**Authors:** Lei Wang, Xili Wang

**Affiliations:** 1School of Computer Science, Shaanxi Normal University, Xi’an 710062, China; 2School of Mathematics and Computer Application, Shangluo University, Shangluo 726000, China; 3Engineering Research Center of Qinling Health Welfare Big Data, Universities of Shaanxi Province, Shangluo 726000, China

**Keywords:** hyperspectral, light detection and ranging, convolutional neural network, graph convolutional network

## Abstract

Deep learning techniques have brought substantial performance gains to remote sensing image classification. Among them, convolutional neural networks (CNN) can extract rich spatial and spectral features from hyperspectral images in a short-range region, whereas graph convolutional networks (GCN) can model middle- and long-range spatial relations (or structural features) between samples on their graph structure. These different features make it possible to classify remote sensing images finely. In addition, hyperspectral images and light detection and ranging (LiDAR) images can provide spatial-spectral information and elevation information of targets on the Earth’s surface, respectively. These multi-source remote sensing data can further improve classification accuracy in complex scenes. This paper proposes a classification method for HS and LiDAR data based on a dual-coupled CNN-GCN structure. The model can be divided into a coupled CNN and a coupled GCN. The former employs a weight-sharing mechanism to structurally fuse and simplify the dual CNN models and extracting the spatial features from HS and LiDAR data. The latter first concatenates the HS and LiDAR data to construct a uniform graph structure. Then, the dual GCN models perform structural fusion by sharing the graph structures and weight matrices of some layers to extract their structural information, respectively. Finally, the final hybrid features are fed into a standard classifier for the pixel-level classification task under a unified feature fusion module. Extensive experiments on two real-world hyperspectral and LiDAR data demonstrate the effectiveness and superiority of the proposed method compared to other state-of-the-art baseline methods, such as two-branch CNN and context CNN. In particular, the overall accuracy (99.11%) on Trento achieves the best classification performance reported so far.

## 1. Introduction

In recent years, the rapid development of multi-sensor technologies has opened up the possibility of observing interest regions on the Earth’s surface from different perspectives. These technologies include mainly synthetic aperture radar (SAR) [1], light detection and ranging (LiDAR) [2], hyperspectral (HS) imagery [3] and multispectral (MS) imagery [4], which all contain several different types of information about the target area. Among them, HS, which is a passive remote sensing technique, samples the reflective part of the electromagnetic spectrum through hyperspectral sensors, resulting in rich and continuous spectral information. This spectral information can range from the visible region (0.4–0.7 μm) to the shortwave infrared region (nearly 2.4 μm) over hundreds of narrow, contiguous spectral channels (typically 10 nm wide). This detailed spectral information makes HS image a valuable data source for remote sensing image classification in complex scenes [5]. However, it is often difficult to obtain satisfactory and fine classification results from single-modal data [6]. For example, there are often many misclassifications in classification tasks of objects consisting of similar materials (e.g., grass, shrubs and trees). Therefore, there is a need to further refine or improve the classification results by combining information from other different sensors. LiDAR, as active remote sensing technology, uses lasers to detect and measure the target area, and obtains exact height and shape information about the scene. Therefore, the comprehensive utilization of HS and LiDAR data has become an active topic in the remote sensing community. Many studies have shown that LiDAR data can be used as an excellent complement to HS data to enhance or improve scene classification performance effectively [7,8]. Due to its significant advantages, multimodal data have also been applied in many other fields, such as crop cover [9] and forest fire management [10].

The fusion methods for multimodal data can currently be divided into two main categories: feature-level fusion and decision-level fusion [11,12]. The former first pre-processes and extracts features from different sources of remote sensing data, then fuses them through feature superposition or reconstruction, and finally sends them to a classifier to complete scene classification. The latter first employs multiple classifiers to classify remote sensing data from multiple sources independently, and finally fuses or integrates these multiple classification results (or decisions) to produce the final classification result. For example, Pedergnana et al. [13] proposed a technique for classifying features extracted from extended attribute profiles (EAPs) on optical and LiDAR images, thus enabling the fusion of spectral, spatial and elevation data. Rasti et al. [14] presented the fusion of the datasets using orthogonal total variance component analysis (OTVCA). It first automatically extracts spatial and elevation information from HS and rasterized LiDAR features, then uses the OTVCA feature fusion method to fuse the extracted spatial and elevation information with the spectral information to generate the final classification map. Rasti et al. [8] put forward a new sparse low-rank based multi-source feature fusion method. It first extracts spatial and elevation information from hyperspectral and LiDAR data using extinction profiles, respectively. Then, low-rank fusion features are estimated from the extracted features using sparse low-rank techniques, which are finally used to generate the final classification map. Sturari et al. [15] introduced a decision-level fusion method using elevation data and multispectral high-resolution imagery, thus enabling the joint use of multispectral, spatial and elevation data. However, most of these studies belong to shallow learning methods, which make it difficult to obtain more refined classification results in complex scenes.

In the last decades, deep learning techniques, represented by convolutional neural networks (CNN), have been able to model high-level feature representations through a hierarchical learning framework and have gradually replaced hand design-based approaches. It has been impressively successful in various fields of computer vision, such as image recognition [16], object detection [17] and natural language processing [18]. In particular, CNN can stack a series of specially designed layers (e.g., the convolutional layers, the batch normalization layer) to form deep models and automatically mine robust and high-level spatial features from a large number of samples, which are typically invariant to most local variations in the data. These deep learning techniques also significantly affect the field of scene classification for multimodal remote sensing data. For example, Xu et al. [19] adopted a two-branch convolutional neural network for extracting features from HS and LiDAR data, respectively, and fed the cascaded features into a standard classifier for pixel-level classification. Li et al. [20] proposed to use deep CNN to learn the features of a pixel pair composed of a central pixel and its surrounding pixels, and then determine the final label based on a voting strategy. Lee et al. [21] developed local contextual interactions by jointly exploiting the local spectral–spatial relationships of adjacent single-pixel vectors, and achieved significant results. In addition, Hong et al. [22] proposed a deep model (EndNet) based on an encoder-decoder structure to extract features from HS and LiDAR data in an unsupervised training manner, and achieved better results. However, deep learning-based methods usually require a large number of labelled samples, which is challenging to meet for remote sensing data.

In recent years, graph convolutional networks (GCN) have developed rapidly and have excelled in the field of unstructured data processing, e.g., social networks [23]. Based on the graph structure, GCN can aggregate and propagate information between any two nodes in a semi-supervised manner, and thus extracting structural features of the data in the middle- and long-range region. This is entirely different from CNN, which extracts neighborhood spatial features in the short-range region [24]. There have been several studies using GCN for the scene classification of HS data. For example, Qin et al. [25] developed a new method of constructing graphs based on combining spatial and spectral features, and improved the ability to classify remote sensing images. Wang et al. [26] designed an architecture to encode the multiple spectral contextual information in a spectral pyramid of multiple embedding spaces with a graph attention mechanism. Wan et al. [27,28] proposed a multi-scale graph convolution operation in the super-pixel technology, which allows for the adjacency matrix to be updated dynamically with network iteration. Besides, miniGCN, proposed by Hong et al. [24], adopted mini-batch learning to train GCN with a fixed scale, combining with local convolutional features for pixel-level classification, and obtaining satisfactory classification results. However, there is currently very little research on the direct use of GCN for multimodal data classification in the remote sensing community.

In summary, the nature of the data captured by different sensors is entirely different, which raises a severe challenge to the effective processing of multi-source remote sensing data [29]. However, it is still a problematic issue to design an effective deep classification model for multimodal data. In this paper, a hyperspectral and LiDAR data classification method based on a dual-coupled CNN-GCN structure (DCCG) is proposed. The model employs two sets of networks with similar structures for structure-level fusion, which can be divided into two parts: a coupled CNN and a coupled GCN. The former utilizes a weight-sharing mechanism to structurally fuse and simplify the dual CNN model and to extract spatial features from hyperspectral and LiDAR images, respectively. The latter first concatenates the HS and LiDAR data for constructing a uniform graph structure. Then, the dual GCN models perform structural fusion by sharing the graph structures and weight matrices of some layers to extract their structural information, respectively. Finally, the final hybrid features are fed into a standard classifier for the pixel-level classification task under a unified feature fusion module. The main contributions can be summarized as follows: A dual-coupled network is proposed for the classification of hyperspectral and LiDAR images. The network adopts a weight-sharing strategy to significantly reduce the number of trainable parameters and alleviate the overfitting of the network. In addition, since CNN and GCN are used, both spatial and structural features can be extracted from hyperspectral and LiDAR data. This brings a richer set of input features to the subsequent classifier and helps to improve the classification performance of the whole network.Several simple and effective multiple feature fusion strategies were developed to effectively utilize the four features extracted from the previous step. These strategies include traditional single fusion strategy and hybrid fusion strategy. Comparative analysis shows that a good feature fusion strategy can effectively improve classification performance.Extensive experiments on two real-world HS and LiDAR data show that the proposed method exhibits significant superiority compared to state-of-the-art baseline methods, such as two-branch CNN and context CNN. In particular, the performance obtained on Trento data is comparable to that of the state-of-the-art methods reported so far.

The remainder of the paper is organized as follows. Section 2 describes the proposed methodology. The dataset and experimental results are presented in Section 3. The proposed method is analyzed and discussed in Section 4. Conclusions are summarized in Section 5.

## 2. Methodology

In this section, the details of our proposed model are described, and how it works is explained.

### 2.1. General Framework of the Proposed Method

In order to effectively extract features from multi-source remote sensing data (HS and LiDAR), we used convolutional neural networks and graph convolutional networks to extract neighborhood spatial features in a short-range region and structural features in a middle- and long-range region, respectively. It should be noted that these two types of features extracted from CNN and GCN are entirely different here, and this has been analyzed and explained in detail in several papers [24]. 

Additionally, we developed two sets of coupled models with similar structures for feature representation or learning by using a weight-sharing mechanism. This may bring two main benefits. On the one hand, it reduces the network parameters, alleviates overfitting and facilitates fast convergence of the model. On the other hand, implicit multi-source data fusion is also partially achieved in the feature extraction phase, as the same set of weights is used to learn the representation of HS and LiDAR data simultaneously. 

So, we propose an HS and LiDAR data classification method based on a dual-coupled CNN-GCN structure (DCCG). Figure 1 illustrates the general framework of our proposed method. The model consists of an input module, a dual-coupled module, a feature fusion module and a classifier module. Note that the proposed dual-coupled module can be divided into two network structures: a coupled convolutional neural network (CCNet) and a coupled graph convolutional network (CGNet), which are described in detail in the subsequent Section 2.2 and Section 2.3. As shown in Figure 1, the inputs to the model are the spectral feature matrix XH∈ℝw×h×D for HS data and the elevation feature matrix XL∈ℝw×h for LiDAR data, respectively. w, h and D denote the original image’s width, height and the number of spectral bands, respectively. The overall output of the network is an end-to-end classification map. In the figure, (CH, CL, GH, GL) are the four features extracted by the dual-coupled module. F represents the fused features processed by the feature fusion module (see Section 2.4 for details).

### 2.2. Feature Extraction on Coupled CNN (CCNet)

It is well known that CNN have demonstrated impressive capabilities in image processing. Similarly, many CNN and their derivatives have been developed to capture the neighborhood spatial features from images in remote sensing classification tasks. These discriminative features will provide the possibility to distinguish target regions in complex scenes.

Figure 2 illustrates the main structure of the coupled CNN model. As can be seen from the figure, the input remote sensing data (i.e., XH or XL) are first sent to a neighborhood patches sampler, which samples and generates a series of neighborhood patches (i.e., three-dimensional spectral cubes PH∈ℝp×p×D or two-dimensional elevation pathes PL∈ℝp×p) from the original data according to a given neighborhood size (p×p). They are then fed into the corresponding CNN model for feature extraction or representation learning. Specifically, the model mainly consists of two almost identical CNN structures for HS and LiDAR data, respectively. The input to the CNN at the top of the figure is PH extracted from the hyperspectral image, while the input to the CNN at the bottom is PL extracted from the LiDAR image. Note that both CNN networks used here contain four convolutional blocks. Each convolutional block contains a similar structure, which consists of a convolutional layer, a batch normalization (BN) layer [30], a MaxPooling layer and a non-linear activation layer. For each CNN network, the number of convolutional kernels in these four convolutional layers is 32, 64, 128, and 128, respectively. The first three convolutional layers use a convolutional kernel size of 3×3, and the last layer is 1×1. CH∈ℝN×128 and CL∈ℝN×128 denote the short-range (or neighborhood) spatial features extracted from the CNN on the HS and LiDAR data, respectively.

Due to the weight-sharing mechanism, the two CNN models share the parameters and settings of the other three convolutional layers, except for the first convolutional layer. This will have the obvious benefit of a large reduction in the number of main parameters of the model. We discuss this in more detail in Section 4.3. Next, the convolution operation rules of the two CNN networks are given as follows:(1)$$H_{j}^{l+1}=f\left(\underset{m}{\sum }H_{m}^{l}W_{j,m}^{l+1}+b_{j,m}^{l+1}\right)$$
where $H_{m}^{l}$ is the mth feature map at the lth layer. $W_{j,m}^{l+1}$ denotes the jth convolution kernel connected with the mth feature map at the (l+1)th layer, and $b_{j,m}^{l+1}$ is the corresponding bias. Here, $H_{}^{0}=P_{H}\;or\;P_{L}$. f(·) denotes the nonlinear activation function, namely ReLU [31]. Obviously, the size of the sampled neighborhood can significantly affect the final classification result. So, this point will be carefully discussed in Section 4.1.

### 2.3. Feature Extraction on Coupled GCN (CGNet)

A large amount of research shows that the neighborhood spatial features extracted by CNN are well suited for processing grid data like images, and have achieved impressive success [32]. However, unlike CNN, GCN can extract middle- and long-range structural features from images based on graph structure. This feature effectively complements CNN and has been used to improve the overall classification accuracy of images in complex scenes [24]. Therefore, naturally, we propose a multi-source remote sensing data classification method based on coupled graph convolutional networks (CGNet). The network attempts to use GCN for multi-source remote sensing data classification tasks and introduces a weight-sharing mechanism for reducing training parameters and preventing overfitting as described previously.

Since the classical GCN suffers from problems, such as difficulty in training on large size graphs, miniGCN [24] is chosen as the backbone to construct the coupled GCN in this paper. It can replace one full-batch training on the full graph with a series of mini-batch training on the sub-graphs.

Similar to CCNet, the proposed CGNet contains two GCN networks with similar structures for extracting (middle- and long-range) structural features from hyperspectral data and LiDAR data, respectively. Figure 3 illustrates the main structure of CGNet. As shown in the figure, the model first integrates the hyperspectral and LiDAR data to form a cascaded raw feature matrix [XH,XL], which is used to construct a unified graph structure G∈ℝN×N. N denotes the number of samples. In this paper, the following radial basis function (RBF) is employed as a similarity measure function for constructing graph structures:(2)Gij=exp(−||xi−xj||σ2)
where xi and xj denote the cascaded HS-LiDAR feature vectors of pixel i and j, respectively. σ is a parameter of RBF that controls the radial range (default: σ = 1). Only the similarities between each pixel and its k-nearest neighbours (default: k = 10) are computed to generate a sparse adjacency matrix.

Next, a joint subgraph sampler is created to allow for mini-batch training on the graph. This sampler can randomly sample from the full graph G at a fixed scale M (M≪N) to generate a series of subgraph sets, A={A1,A2,⋯,As}, s=1,2,⋯,⎡N/M⎤. As∈ℝM×M. As shown in the upper part of Figure 3, the generated subgraph set A and all spectral features VH∈ℝN×D for each pixel are then fed into the GCN to complete the structural feature extraction for the HS data. Correspondingly, the same subgraph set A and the elevation features VL∈ℝN for each pixel are also be fed into the GCN to perform the structural feature extraction for the LiDAR data, as shown in the lower part of Figure 3.

Note that the two GCN networks have nearly identical network structures. Each GCN consists of just two hidden layers with 32 and 128 units, respectively (too many hidden layers in a GCN would lead to overly smooth results [23]). Due to the difference in input layers, two GCN networks can only share their trainable weight matrix of the second hidden layer according to the weight-sharing strategy. Here, the rules for propagation between the layers of these two GCN networks are presented:(3)H˜sl+1=f(D˜−12A˜sD˜−12H˜slWl+1+bl+1)
(4)Hl+1=[H˜1l+1,⋯,H˜sl+1,⋯,H˜⎡N/M⎤l+1]
where s is not only the sth subgraph but also the sth batch in the network training process. A˜s=As+IM, D˜i,i=∑jA˜i,j. As is the adjacency matrix on the sth subgraph. IM is the identity matrix with M nodes. Wl+1 and bl+1 represent the learnable weight parameter and bias of the (l+1)th hidden layer, respectively. H˜sl+1 denotes the outputs of the sth subgraph in the lth hidden layer. Hl+1 denotes the total output of the lth layer on the whole graph, where H0=VH or VL. f(·) represents the ReLU activation function. GH∈ℝN×128 and GL∈ℝN×128 denote the middle- and long-range structural features extracted by the GCN on the HS and LiDAR data, respectively.

### 2.4. Training and Testing of the Model

After CCNet and CGNet processing, we can extract four features (CH, CL, GH and GL) from the hyperspectral and LiDAR data. An effective data fusion (i.e., fusion module) of these features is required for better classification before feeding into the classifier. Here, for simplicity, the fusion of the multi-source data is accomplished in a cascade fashion as follows:(5)F=[CH,CL,GH,GL]
where F∈ℝN×512 denotes the fused features. Note that more comparisons of fusion strategies will be discussed in Section 4.2.

Next, the fused features are fed into a standard softmax classifier, consisting of two fully connected layers containing 128 and C units, respectively. The final output of the whole network can be computed as:(6)$$\hat{y}=softmax\left(W_{2}f\left(W_{1}F+b_{1}\right)+b_{2}\right)$$
where $W_{1}$ and $W_{2}$ denote the weight matrix of the two fully connected layers, respectively. $b_{1}$ and $b_{2}$ are their respective bias. 

The loss of the whole model can be defined by the cross-entropy loss function as follows:(7)$$L=-\frac{1}{N}\overset{N}{\underset{i=1}{\sum }}\left(y_{i}log{\hat{y}}_{i}+\left(1-y_{i}\right)log\left(1-{\hat{y}}_{i}\right)\right)$$
where $y_{i}$ and ${\hat{y}}_{i}$ indicate the actual class label and the predicted label of pixel i, respectively. Using Adam [33], back propagation and parameter updates are performed on the whole model.

## 3. Experiments

To evaluate the effectiveness of the proposed method, we compare DCCG with several other methods, such as CNN [31], miniGCN [24], CCNet (see Section 2.2) and CGNet (see Section 2.3). Four metrics, such as per-class accuracy, overall accuracy (OA), average accuracy (AA) and kappa coefficient (κ), are used to measure classification accuracy quantitatively. Classification maps are adopted to evaluate the classification results qualitatively. Furthermore, to demonstrate the superiority of the proposed method, several state-of-the-art multimodal remote sensing classification models are compared in Section 4.4.

### 3.1. Data Description

All experiments in this paper were conducted on two real hyperspectral and LiDAR data, i.e., Trento and Houston. 

Trento Data: The data are composed of the HSI and LiDAR images that were acquired over the area from the city of Trento, Italy. This scene consists of 166 × 600 pixels with 1 m spatial resolution. The HSI scene consists of 63 spectral bands with wavelength ranging from 0.42 to 0.99 μm including six classes [22]. Figure 4 visualizes the HS and LiDAR as well as the locations of training and testing samples on Trento data. Table 1 lists the information about the number of samples in different classes on Trento data.Houston Data: The data are composed of the HSI and LiDAR images that were captured in 2012 by an airborne sensor over the area of University of Houston campus and the neighboring urban area. This scene consists of 349 × 1905 pixels with 2.5 m spatial resolution. The HSI scene consists of 144 spectral bands with wavelength ranging from 0.38 to 1.05 μm including 15 classes [34]. Figure 5 visualizes the HS and LiDAR as well as the locations of training and testing samples on Houston data. Table 2 lists the information about the number of samples in different classes on Houston data.

### 3.2. Experimental Setup

According to the properties of the methods, these classification methods involved in this experiment can be divided into two main categories: single model-based methods and hybrid model-based methods. The former includes CNN-H, CNN-L, miniGCN-H and miniGCN-L, while the latter includes CCNet-HL, CGNet-HL, DCCG-H, DCCG-L and DCCG-HL. After the symbol ‘-’, the letters ‘H’ and ‘L’ indicate HS and LiDAR data, respectively. Note that the network structure of the CNN and miniGCN models in the comparison experiments is the same as described in Section 2.2 and Section 2.3 of this paper. The maximum number of epochs, learning rate and batch size were set to 200, 0.001 and 32, respectively. Here, for comparison purposes, the neighborhood size *p* is set to 11 on both Trento data and Houston data. More neighborhood size settings will be discussed in Section 4.1. In particular, to facilitate computation and reduce data redundancy, as with most multimodal classification methods, we first performed PCA dimensionality reduction on the hyperspectral data in both experimental datasets before model training. Here, the first 20 dimensional features were empirically taken as model input before subsequent classification [35,36]. All deep models were implemented in the TensorFlow 1.5 framework. All experiments were carried out in the same hardware environment, i.e., Intel i5-12400, 16 GB memory and no GPU devices. 

### 3.3. Results and Analysis of Trento Data

Table 3 lists the quantitative classification results obtained by the different methods on Trento data. From the table, some remarkable facts can be observed. Firstly, for all models using only single source (HS or LiDAR) data, the performance using only hyperspectral data is significantly higher than that of only LiDAR data. For example, miniGCN-H shows a 23.91% improvement on OA over miniGCN-L. This is mainly because hyperspectral images contain rich spectral information and fine neighborhood spatial information compared to LiDAR data, which provide many distinguishable features for effective classification. Furthermore, we observed that CNN significantly outperformed miniGCN in all single model-based methods. For instance, the overall accuracy was improved from 81.79% for miniGCN-H to 93.81% for CNN-H on hyperspectral data by 12.02%. One of the possible reasons for this phenomenon is that the CNN-extracted short-range (or neighborhood) features are more helpful for pixel classification than the GCN-extracted middle- and long-range structural features. Thirdly, the introduction of multiple source data and hybrid structures significantly improved the classification performance of the models compared to the single-source data and single-model approaches. For instance, CCNet-HL and CGNet-HL improved by 3.15% and 6.08% on OA, respectively, over their single-model approaches. This is partly due to the inclusion of multi-source data, making the classifier’s input more multi-source and abundant, while enhancing the dimensionality of the distinguishable features. More importantly, on the other hand, the adoption of a weight-sharing-based coupling structure allows for a significant reduction in the model parameters, which alleviates overfitting and improves the model’s generalization ability. Finally, our proposed method was found to achieve the best performance of all the methods in the experiment. It not only benefits from multi-source data and the coupling structure based on weight sharing, but also because the neighborhood spatial information and the middle- and long-range structural information are considered simultaneously in the model, resulting in the complementarity of information at the feature level. Specifically, DCCG-HL consists of CCNet-HL and CGNet-HL, but its classification performance is remarkably higher than the latter two by 4.65% and 6.85%, respectively. This also demonstrates the effectiveness of the proposed method from the side. 

Figure 6 shows the full classification maps of the different methods on the Trento data. As can be seen, the inclusion of multiple-sources data significantly improves the classification results. Our proposed method retains the boundaries between lands while having smoother regions. Figure 7 illustrates the loss curves obtained by different methods for the Trento dataset. Similarly, our DCCG-HL successfully suppresses the overfitting and achieves the best test results.

### 3.4. Results and Analysis of the Houston Data

Table 4 reports the quantitative classification results obtained by the different methods on Houston data. From the table, a similar fact to Trento can be observed. Our proposed method still achieves the best classification results and outperforms the second place (CCNet-HL) by 1.99% on OA. It also approaches 100% accuracy in classes 1, 3, 7 and 15. In addition, an interesting phenomenon was observed. Specifically, in class 13, the classification accuracy of CCNet-HL and CGNet-HL was 74.46% and 69.92%, respectively. While our proposed DCCG-HL combined these two models, its OA improved by 15.32% to 89.78%. The main reason for this phenomenon is that the proposed method integrates well the neighborhood spatial features and the middle- and long-range structural features, achieving complementarity between the different features. 

Figure 8 exhibits the full classification maps obtained by the different classification methods on the Houston data. On one hand, CNN-based methods (e.g., Figure 8a,e) can utilize the extracted neighborhood spatial features to make the classified region very smooth, but the edges and details are often ignored (e.g., roads). GCN-based methods (e.g., Figure 8c,f) can employ middle- and long-range structural features to retain more detailed information, but with more noise in the classified regions. This reflects the classification tendencies of the different classification methods. On the other hand, hyperspectral-based methods (e.g., Figure 8a,c,g) clearly provide more detailed information about the target using their rich spectral bands, which is very beneficial for the classification task. However, it has difficulty in solving the problem of classifying similar objects (e.g., grass and shrubs). LiDAR-based methods (e.g., Figure 8d) use elevation information to easily distinguish between targets with different heights, but targets with the same height are difficult to classify. This is a limitation of single-sensor classification. In contrast, our proposed DCCG-HL integrates both multi-sensor and multi-model techniques to effectively perform the classification task in complex scenes. Specifically, it can smooth the classified region while retain more classification detail information (e.g., roads) between regions. Figure 9 displays the loss curves obtained by different methods for the Houston dataset. Similar to Trento data, our proposed DCCG-HL method is smoother throughout the training and testing process and successfully suppresses the overfitting.

## 4. Discussion

### 4.1. Effect of Different Neighborhood Sizes on Classification Results

In the proposed model, one parameter (i.e., neighborhood size p×p) needs to be determined in advance before training. We fixed the basic structure and main parameters of the network, picked the neighborhood size p from the candidate set {5,7,9,11,13,15,17,19,21} in turn, and tested the overall accuracy variation of the network. Figure 10 illustrates the effect of different neighborhood sizes on the classification accuracy of the Trento and Houston data, respectively. A similar phenomenon can be observed in the figure, where the OA first rises significantly and then slowly decreases as the neighborhood size increases. Specifically, the model achieves the highest classification accuracy of 99.06% on Trento data when p=11. Correspondingly, the model also achieves the highest classification accuracy of 93.79% on Houston data when p=17.

### 4.2. Effect of Different Fusion Strategies on Classification Results

The effective fusion of features for multi-source data classification is a key step. Here we discuss the impact of different fusion strategies on classification results. These strategies can be divided into traditional single-based fusion strategies and hybrid-based fusion strategies. Specifically, the former contains additive fusion (DCCG-A), pixel-wise multiplicative fusion (DCCG-M) and cascade fusion (DCCG-C). As for the latter, similar to Equation (5), we give definitions of three hybrid-based fusion strategies: DCCG-1 ($\left[C_{h}+C_{l},G_{h}+G_{l}\right]$), DCCG-2 ($\left[C_{h}+G_{h},C_{l}+G_{l}\right]$) and DCCG-3 ($\left[C_{h}+G_{l},C_{l}+G_{h}\right]$).

Table 5 reports the effect of different fusion strategies on the classification results. As can be seen from the table, for the Houston data, the cascade fusion method (DCCG-C) slightly outperformed the DCCG-2 method and achieved the best OA. In contrast, for the Trento data, the DCCG-2 strategy slightly outperformed DCCG-1. 

### 4.3. Analysis of the Number of Parameters

To demonstrate the effectiveness of the coupled structure, we counted the total number of parameters and the overall accuracy of DCCG before and after using the coupled structure (see Table 6). As can be seen from the table, the total number of parameters in the model was reduced from 301,936 to 188,848 on the Trento data and from 598,777 to 338,105 on the Houston data after adopting the weight-sharing module (i.e., coupled structure). In other words, the total parameters were decreased by 37% and 44% on the Trento and Houston data, respectively. Fortunately, the OA on these two datasets was slightly increased by 0.55% and 0.62%, respectively. This demonstrates that the weight-sharing module can both reduce the model parameters and improve the classification results in complex scenes.

### 4.4. Comparison with Other State-of-the-Art Methods

To verify the superiority of the proposed method, Table 7 compares several state-of-the-art multimodal classification methods: EndNet [22], MML(FC) [37], CNN-PPF(H + L) [20], Two-Branch CNN(H + L) [19], Context CNN(H + L) [21] and PToP CNN [38]. It should be noted that, except EndNet and MML(FC), the authors did not provide the original code for the implementation. Therefore, the classification results for CNN-PPF (H + L), Two-Branch CNN (H + L), Context CNN (H + L) and PToP CNN are taken directly from the best results in the relevant literature. For MML(FC), the original one-dimensional LiDAR is used here for testing as the authors do not provide a specific pre-processing procedure for Trento’s LiDAR data. The superiority of the proposed method can be seen in the table, which outperforms current state-of-the-art classification methods such as Two-Branch CNN and Context CNN on both Trento data and Houston data. Furthermore, we observe that the classification accuracy (99.11%) of the Trento data achieves the best classification performance reported so far.

### 4.5. Analysis on the Computation Cost

In order to evaluate the computation cost of the different methods, Table 8 and Table 9 report the computational time on the Trento and Houston data, respectively. Unfortunately, for the same reason as mentioned in Section 4.4, the authors’ source code was not available and thus some advanced methods, i.e., CNN-PPF (H + L), Two-Branch CNN (H + L), Context CNN (H + L) and PToP CNN, could not be evaluated in the same experimental setting in terms of computational time. Furthermore, it should be noted that all experiments were performed on the Tensorflow framework and without using any GPU acceleration techniques.

From the table, it can be observed that all single-source models take less time to train than their corresponding two-source models. This is mainly due to the fact that the single-source models deal with less data. In contrast, the two-source models need to process more remote sensing data from more sources and thus require more training time. However, once all models have been trained, they are tested very efficiently. For DCCG-HL, the introduction of weight sharing did not significantly increase the training and testing time.

## 5. Conclusions

This paper proposed a dual-coupled CNN-GCN-based classification method for hyperspectral and LiDAR data. The method can be divided into two sets of structurally similar networks: a coupled CNN and a coupled GCN. Such two networks can be employed to extract spatial-spectral features in a short-range region and structural features in a middle- and long-range region from hyperspectral and LiDAR data, respectively. Furthermore, these two classes of features are proven to achieve feature complementarity and performance enhancement in multi-source remote sensing data classification tasks. The coupling module using weight sharing makes the network with a substantially smaller number of parameters, alleviating overfitting and improving the model’s generalization performance. The effects of different neighborhood sizes and feature fusion strategies on the final classification results of the model are discussed in detail. Extensive experiments on two real-world hyperspectral and LiDAR images demonstrate the clear superiority of the proposed method compared to other state-of-the-art methods for multimodal data classification. In future research, we will continue to explore more efficient feature extraction methods and multimodal fusion methods in remote sensing data classification.

## Figures and Tables

**Figure 1 sensors-22-05735-f001:**
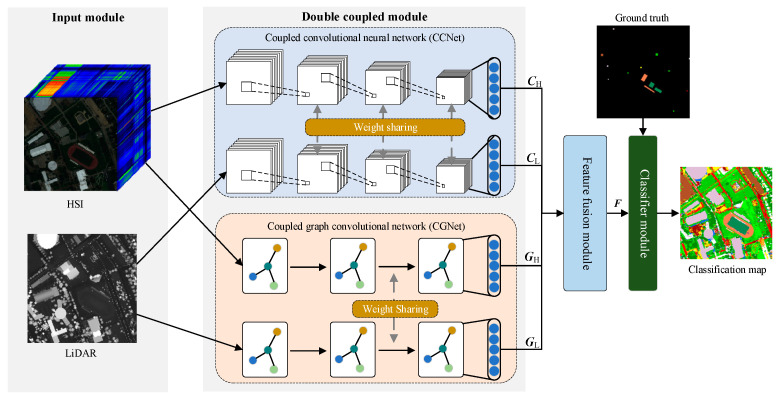
A general framework of proposed DCCG.

**Figure 2 sensors-22-05735-f002:**
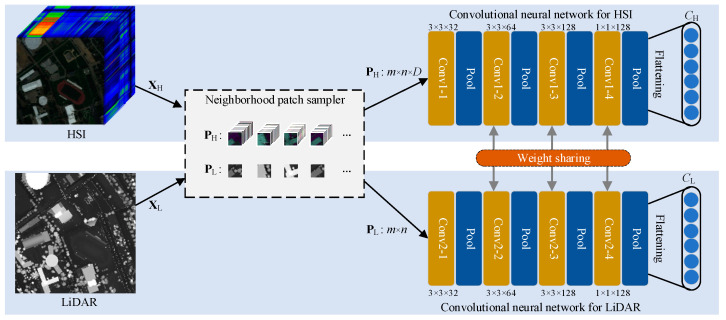
Illustration of the structure of CCNet. Conv and Pool denote the convolution and maximum pooling operations, respectively. p is the given neighborhood size.

**Figure 3 sensors-22-05735-f003:**
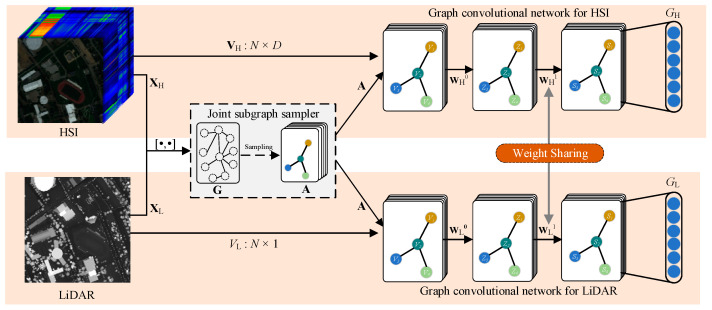
Illustration of the structure of CGNet. [⋅,⋅] indicates cascade operations.

**Figure 4 sensors-22-05735-f004:**
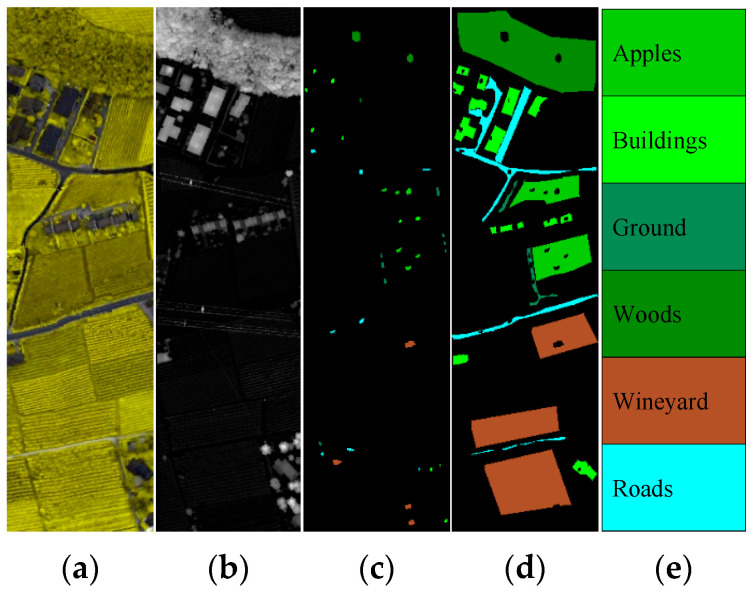
Visualization of Trento dataset: (**a**) false-color image (using 50, 45 and 5 bands as R, G and B, respectively); (**b**) LiDAR image; (**c**) training samples; (**d**) test samples; (**e**) color code.

**Figure 5 sensors-22-05735-f005:**
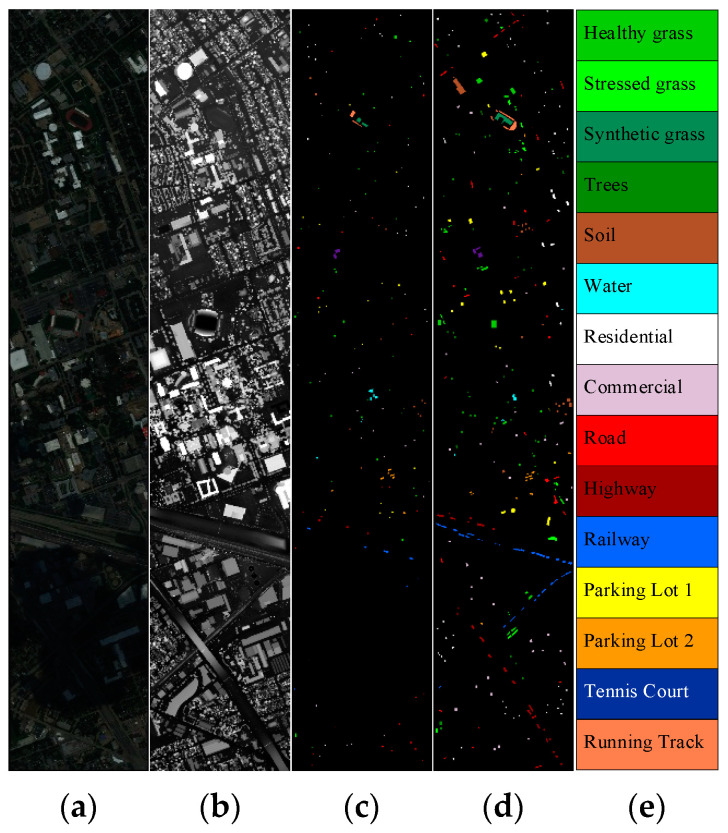
Visualization of Houston dataset: (**a**) false-color image (using 64, 43 and 22 bands as R, G and B, respectively); (**b**) LiDAR image; (**c**) training samples; (**d**) test samples; (**e**) color code.

**Figure 6 sensors-22-05735-f006:**
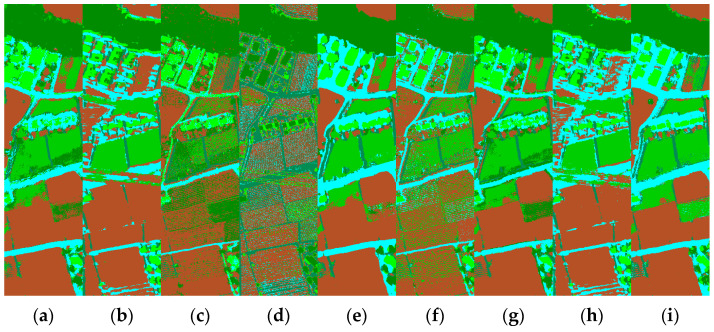
Classification maps obtained by different methods for the Trento dataset. (**a**) CNN-H; (**b**) CNN-L; (**c**) miniGCN-H; (**d**) miniGCN-L; (**e**) CCNet-HL; (**f**) CGNet-HL; (**g**) DCCGNet-H; (**h**) DCCGNet-L; (**i**) DCCG-HL.

**Figure 7 sensors-22-05735-f007:**
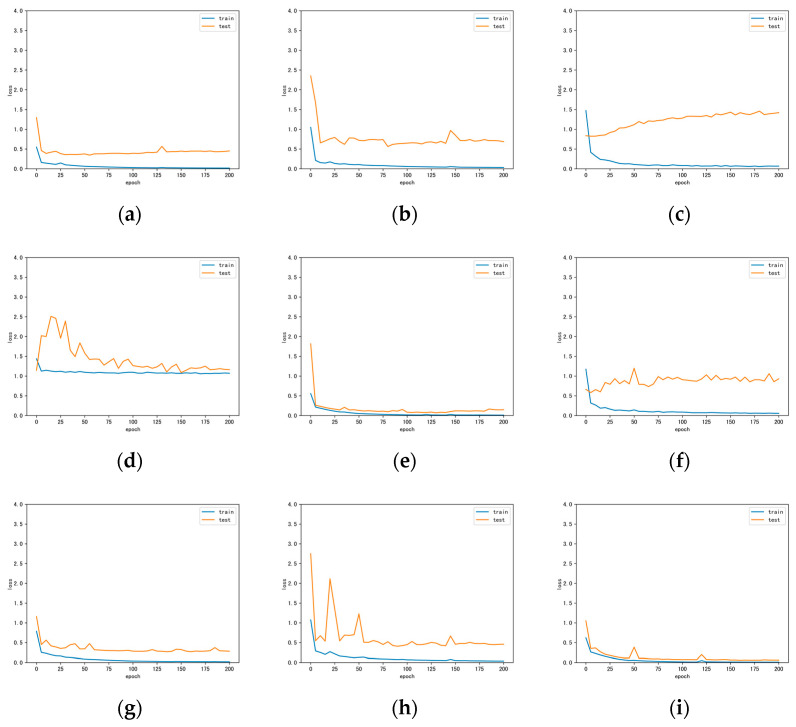
Loss curves obtained by different methods for the Trento dataset. (**a**) CNN-H; (**b**) CNN-L; (**c**) miniGCN-H; (**d**) miniGCN-L; (**e**) CCNet-HL; (**f**) CGNet-HL; (**g**) DCCGNet-H; (**h**) DCCGNet-L; (**i**) DCCG-HL.

**Figure 8 sensors-22-05735-f008:**
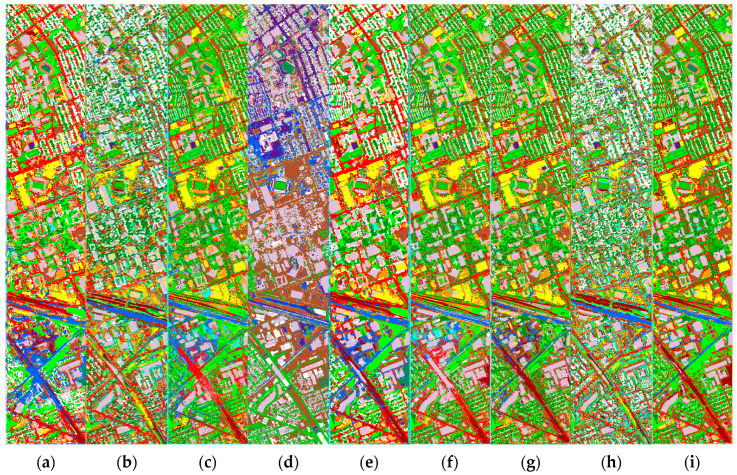
Classification maps obtained by different methods for the Houston dataset. (**a**) CNN-H; (**b**) CNN-L; (**c**) miniGCN-H; (**d**) miniGCN-L; (**e**) CCNet-HL; (**f**) CGNet-HL; (**g**) DCCGNet-H; (**h**) DCCGNet-L; (**i**) DCCG-HL.

**Figure 9 sensors-22-05735-f009:**
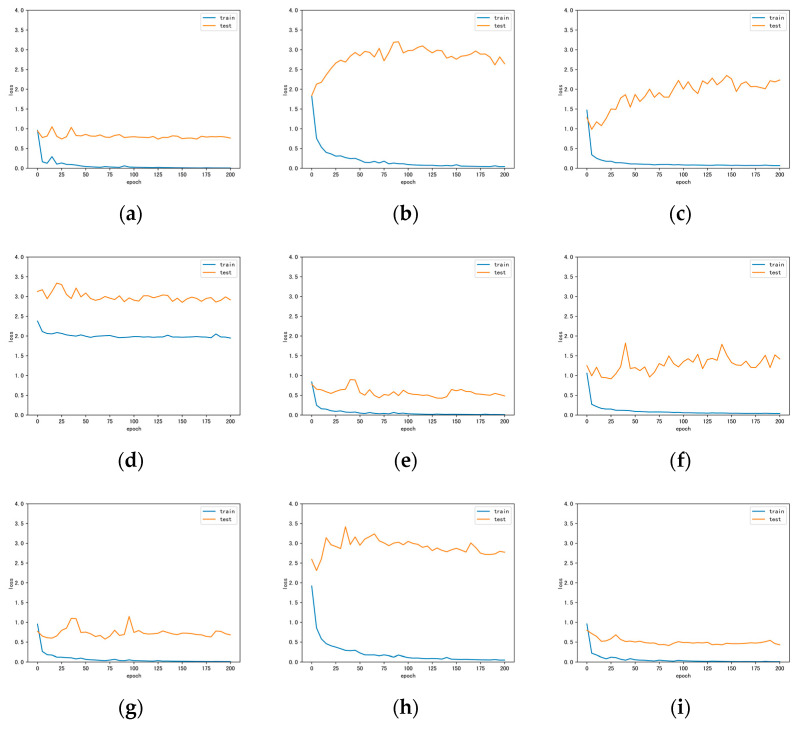
Loss curves obtained by different methods for the Houston dataset. (**a**) CNN-H; (**b**) CNN-L; (**c**) miniGCN-H; (**d**) miniGCN-L; (**e**) CCNet-HL; (**f**) CGNet-HL; (**g**) DCCGNet-H; (**h**) DCCGNet-L; (**i**) DCCG-HL.

**Figure 10 sensors-22-05735-f010:**
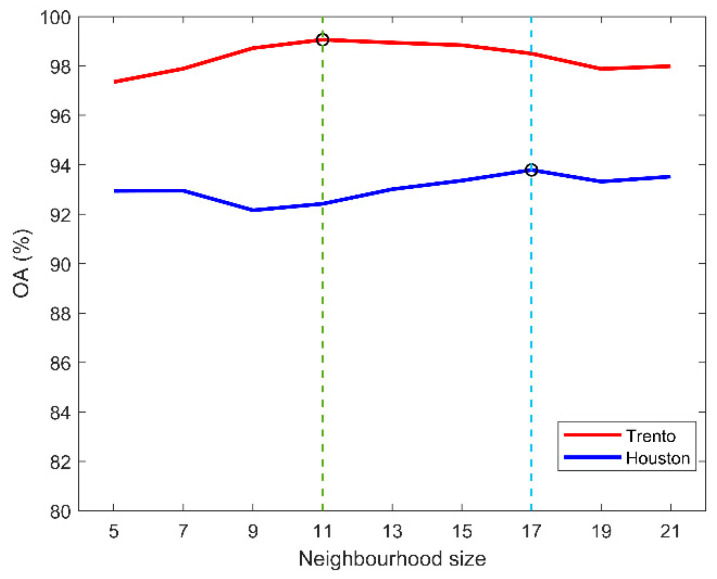
Effect of different neighborhood sizes on OA.

**Table 1 sensors-22-05735-t001:** List of the number of training and test samples for each class in Trento datasets.

No.	Class Name	Training	Test
1	Apples	129	3905
2	Buildings	125	2778
3	Ground	105	374
4	Woods	154	8969
5	Wineyard	184	10,317
6	Roads	122	3052
-	Total	819	29,395

**Table 2 sensors-22-05735-t002:** List of the number of training and test samples for each class in Houston datasets.

No.	Class Name	Training	Test
1	Healthy grass	198	1053
2	Stressed grass	190	1064
3	Synthetic grass	192	505
4	Trees	188	1056
5	Soil	186	1056
6	Water	182	143
7	Residential	196	1072
8	Commercial	191	1053
9	Road	193	1059
10	Highway	191	1036
11	Railway	181	1054
12	Parking Lot 1	192	1041
13	Parking Lot 2	184	285
14	Tennis Court	181	247
15	Running Track	187	473
-	Total	2832	12,197

**Table 3 sensors-22-05735-t003:** Per-class accuracy, OA, AA and κ obtained by different methods on Trento data. The bold text in each row indicates the best classification result.

No.	CNN-H	CNN-L	miniGCN-H	miniGCN-L	CCNet-HL	CGNet-HL	DCCG-H	DCCG-L	DCCG-HL
1	93.17	88.48	73.24	6.75	98.23	56.45	98.07	87.24	**99.61**
2	74.20	88.61	69.82	22.78	85.80	**96.31**	73.71	92.13	96.20
3	**90.00**	27.05	50.42	4.75	60.71	77.20	74.89	27.68	89.74
4	98.57	95.28	90.62	76.26	**100.0**	99.35	99.33	96.18	99.99
5	98.56	93.78	80.98	72.54	99.03	90.97	98.08	96.70	**99.48**
6	83.82	93.91	87.03	6.85	95.98	97.53	85.90	86.40	**98.10**
OA	93.81	91.22	81.79	57.88	96.96	87.87	94.41	92.21	**99.06**
AA	89.72	81.19	75.35	31.65	89.95	86.30	88.33	81.05	**97.19**
κ	91.69	88.18	75.40	43.11	95.92	84.02	92.50	89.58	**98.75**

**Table 4 sensors-22-05735-t004:** Per-class accuracy, OA, AA and κ obtained by different methods on Houston data. The bold text in each row indicates the best classification result.

No.	CNN-H	CNN-L	miniGCN-H	miniGCN-L	CCNet-HL	CGNet-HL	DCCG-H	DCCG-L	DCCG-HL
1	96.73	49.96	96.82	40.07	99.20	99.11	**99.77**	46.71	99.32
2	80.51	24.85	92.91	0.00	86.00	92.03	**92.96**	24.97	83.65
3	90.95	45.57	**100.00**	33.29	93.01	98.24	98.83	47.34	99.80
4	96.64	83.04	93.64	13.92	**97.63**	93.55	87.44	82.82	97.21
5	86.88	38.32	96.21	5.79	93.74	97.13	91.56	31.68	**97.77**
6	82.98	10.61	42.90	1.25	82.42	68.66	**91.61**	12.14	86.79
7	89.15	66.67	98.92	39.08	93.99	98.71	95.85	66.74	**99.17**
8	82.90	67.13	76.57	26.98	82.86	68.36	84.30	67.85	**91.43**
9	82.78	41.68	62.32	25.00	**86.13**	72.82	84.07	52.24	81.75
10	77.71	80.08	80.00	0.00	**91.83**	80.90	84.99	74.68	88.88
11	70.01	88.75	85.96	45.93	92.81	94.22	87.70	89.29	**97.13**
12	78.17	30.95	80.39	0.00	84.77	80.39	82.02	36.75	**85.83**
13	75.54	44.93	73.68	3.63	74.46	69.92	69.48	52.44	**89.78**
14	87.02	42.22	90.77	12.59	**100.00**	96.08	89.17	31.68	96.86
15	82.65	28.72	99.36	0.00	93.22	99.79	**100.00**	22.50	99.79
OA	83.79	52.50	85.61	23.24	90.43	87.42	89.23	52.39	**92.42**
AA	84.04	49.57	84.70	16.50	90.14	87.33	89.32	49.32	**93.01**
κ	82.39	48.60	84.39	17.90	89.61	86.35	88.31	48.52	**91.77**

**Table 5 sensors-22-05735-t005:** Effect of different fusion strategies on OA (%).

Datasets	DCCG-A	DCCG-M	DCCG-C	DCCG-1	DCCG-2	DCCG-3
Trento	98.74	98.83	99.06	99.08	99.11	98.91
Houston	91.9	92.7	93.79	93.13	93.29	92.86

**Table 6 sensors-22-05735-t006:** Changes in the number of model parameters by the weight-sharing module.

Datasets	DCCG without Weight Sharing		DCCG with Weight Sharing	
	No. of Parameters	OA (%)	No. of Parameters	OA (%)
Trento	301,936	98.56	188,848	99.11
Houston	598,777	93.17	338,105	93.79

**Table 7 sensors-22-05735-t007:** Comparison with state-of-the-art methods on OA (%).

Datasets	EndNet	MML(FC)	CNN-PPF (H + L)	Two-Branch CNN (H + L)	Context CNN (H + L)	PToP CNN	Our DCCG
Trento	92.34	91.62	94.76	97.92	96.11	98.34	99.11
Houston	88.43	89.69	83.33	87.98	86.90	92.48	93.79

**Table 8 sensors-22-05735-t008:** Computation time (seconds) of different methods on the Trento data.

**Time (s)**	**CNN-H**	**CNN-L**	**miniGCN-H**	**miniGCN-L**	**CCNet-HL**	**CGNet-HL**
Train	192.72	147.72	388.81	378.38	1062.55	616.24
Test	4.88	2.33	6.46	6.34	8.03	8.65
**Time (s)**	**DCCG-H**	**DCCG-L**	**DCCG-HL (without Weight Sharing)**	**Our DCCG-HL**	**EndNet**	**MML(FC)**
Train	935.85	729.92	1160.20	1198.10	28.71	87.87
Test	12.69	9.00	14.34	14.70	0.19	0.34

**Table 9 sensors-22-05735-t009:** Computation time (seconds) of different methods on the Houston data.

**Time (s)**	**CNN-H**	**CNN-L**	**miniGCN-H**	**miniGCN-L**	**CCNet-HL**	**CGNet-HL**
Train	321.21	267.44	132.76	125.36	944.73	200.20
Test	2.55	0.87	2.94	2.74	3.77	3.64
**Time (s)**	**DCCG-H**	**DCCG-L**	**DCCG-HL (without Weight Sharing)**	**Our DCCG-HL**	**EndNet**	**MML(FC)**
Train	622.51	589.55	955.83	1003.72	27.20	100.49
Test	5.28	3.76	6.40	6.59	0.07	0.15

## Data Availability

The data used to support the findings of this study are available from the co-first author upon request.

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
