# Peer review of "Dual-Coupled CNN-GCN-Based Classification for Hyperspectral and LiDAR Data"

_sensors, 2022, doi:10.3390/s22155735_

Round 1

Reviewer 1 Report

The authors propose a dual-coupled CNN-GCN structure to deal with classification in HS and LiDAR data.

The structure presented by the authors is interesting, however, I have some questions about the choices:

- Instead of weigth sharing, would it be possible to perform some kind of two-step training procedure, using the previous trained weights of one network as the initial weights of the other? Could this improve the computational time without losing accuracy performance?

- More tests about overfitting would be appreciated;

- I would also be interested in the training times when compared to the other presented benchmark networks. Does this weigth sharing technique makes the overall convergence slower?

Reviewer 2 Report

Dear Authors, 

I have reviewed the paper entitled "Dual-Coupled CNN-GCN-Based Classification for Hyperspectral and LiDAR Data". The paper presents a classification method based on Hyperspectral and LiDAR data. In my opinion the paper is interesting.

Kind regards.

1. I see the lack of information on testing algorithms in empirical environments. It is known that some objects are easier and some more difficult to classify. By adding a few references for the type of test fields, in my opinion is a good addition to the article.

2. Under what conditions were the measurements performed ? How did they influence the results ? Did the authors manually check the results for correctness ?

3. If the Authors show visualizations of the research objects, it should be more professional - such as adding a scale, a north sign and ensuring better legibility of the Figures.

These are small comments. For me, all the paper is legible and understandable.

Reviewer 3 Report

The authors investigated dual-coupled CNN-GCN-based classification for hyperspectral and LiDAR data. They employ a weight-sharing mechanism to fuse dual CNN model, concatenate the HS and LiDAR data to construct a uniform graph structure, and then feed the final hybrid features into a standard classifier for the pixel-level classification task. The effectiveness of the method was demonstrated by two real-world hyperspectral and LiDAR data. I recommend this manuscript to be accepted in present form.

Reviewer 4 Report

# Summary

This paper proposed a [new] engineered solution exploring CNN and GCN to solve the problem of HSI classification. In order to enrich the feature set, the authors proposed data fusion with LiDAR. This would potentially help classification by extending the HSI features (spatial+spectral), with shape information (such height from the ground, volume of objects, others).
Despite indications of overfitting, the results seems appropriate and still relevant to the field.

# Criticism

The paper would benefit with more references on GCN and CNN, specifically to Lidar-only, HSI-only and data fusion. Also, the paper lacks better comparison with existing work that applies the similar ideas, such as “coupled models”, weight-sharing and data fusion. I also believe this paper could mention existing work using other data fusion such as: HSI with SAR and RGB with SAR.The training regime could be better described. It is not clear how the CNN and GCN runs on the HSI, i.e., how the filters are applied throughout the spectral dimension: is it band by band? Is there a pre-selection of bands for feature extraction with CNN and GCN? The high dimensionality of HSI (spectral-wise) brings extra challenges to the training, wrt to. the minima search of CNNs and GCNs. The training sets are noticeable small, therefore one can assume the models are overfitting to the specific datasets, therefore less generalizable.
Even with the coupled models/weight sharing, the overfitting issue is likely preponderant. If not, the design of experiments should accommodate experiments to bring evidence of non-overfitted models.This research could explore transfer learning to alleviate the issue of lack of ground-truth for training. If this can’t be explored or it is out of scope, it would be beneficial to describe, explain, and discuss limitations of the current method with the datasets used.The methods are well described and the authors made it easy to understand.
However, some minor details could be better covered such as the weight-sharing mechanism. Lack explanations on the training regime and hyperparameter tuning, for better reproducibility.  The availability of source code would also help.Different metrics would also improve understanding of the classification performance.

# Comments and Suggestions for Authors

Line 42 minor typos: the wavelength units must be fixed. It was supposed to be 0.4-0.7 micrometers and 2.4 micrometers.
(Currently shows meters is incorrect)

Round 2

Reviewer 1 Report

The authors addressed all my issues.